# Normal & reversed spin mobility in a diradical by electron-vibration coupling

Yi Shen[1], Guodong Xue[1], Yasi Dai [2,3], Sergio Moles Quintero [4], Hanjiao Chen[5], Dongsheng Wang[1], Fang Miao[1✉], Fabrizia Negri [2,3✉], Yonghao Zheng [1✉] & Juan Casado [4✉]

$\pi$—conjugated radicals have great promise for use in organic spintronics, however, the mechanisms of spin relaxation and mobility related to radical structural flexibility remain unexplored. Here, we describe a dumbbell shape azobenzene diradical and correlate its solid-state flexibility with spin relaxation and mobility. We employ a combination of X-ray diffraction and Raman spectroscopy to determine the molecular changes with temperature. Heating leads to: i) a modulation of the spin distribution; and ii) a "normal" quinoidal → aromatic transformation at low temperatures driven by the intramolecular rotational vibrations of the azobenzene core and a "reversed" aromatic → quinoidal change at high temperatures activated by an azobenzene bicycle pedal motion amplified by anisotropic intermolecular interactions. Thermal excitation of these vibrational states modulates the diradical electronic and spin structures featuring vibronic coupling mechanisms that might be relevant for future design of high spin organic molecules with tunable magnetic properties for solid state spintronics.

[1] School of Optoelectronic Science and Engineering, University of Electronic Science and Technology of China (UESTC), Chengdu 610054, People's Republic of China. [2] Università di Bologna, Dipartimento di Chimica 'Giacomo Ciamician', Via F. Selmi, 2, 40126 Bologna, Italy. [3] INSTM, UdR, Bologna, Italy. [4] Department of Physical Chemistry, University of Málaga, Campus de Teatinos s/n, Málaga 29071, Spain. [5] Analytical & Testing Center, Sichuan University, Chengdu 610064, People's Republic of China. ✉email: miaofang@uestc.edu.cn; fabrizia.negri@unibo.it; zhengyonghao@uestc.edu.cn; casado@uma.es

The spin is an intrinsic property of matter in the atomic world playing a decisive role in the physics of metals and semiconductors as well as in the electro-optical properties of organic π-conjugated materials. In the latter, π-organic molecules, the microscopic mechanisms of spin mobility/delocalization and spin-relaxation (i.e., hyperfine, spin-orbit, or spin-lattice couplings) are poorly understood given their inherent weaknesses[1–3]. On the way to uncover the fundamental properties of the spin in the organic matter, an important feature is the description of the connections between structural flexibility of π-conjugated molecules and spin delocalization and dynamics in the flexible backbone.

Diradicals have always attracted the attention of chemists owing to their unique role in understanding the fundamental nature of chemical bonds and of reactivity[4–8]. In the last decade, the class of diradicals based on open-shell π-conjugated structures have become of interest in various organic electronic and magnetic applications[9–11]. Importantly, thermal modulation of the diradical character might be an important venue for the designs of new flexible organic magnetic materials, an aspect that remains scarcely explored[12–15].

The azobenzene moiety has been shown to be a solid-state building scaffold[16] able to bring intramolecular torsional mobility to their crystals consisting in non-translational displacements such as the so-called bicycle pedal motion associated to the inversion of the NN double bond orientation relative to the lateral benzenes, a unique feature first described in carotenoids (e.g., retinal)[17] associated with the sequence of double and single bonds of the conjugated path (Fig. 1).

Pursuing to implement the azobenzene moiety with magnetic activity, we here design a diradical molecule through a straightforward coupling reaction between azobenzene with two lateral phenoxyl units (conjugated bis(phenyloxy) azobenzene diradical (CAR) in Fig. 1. The dual role of the azobenzene providing the electronic correlation between the two radicals and a flexible NN bridge produces a unique magnetic solid. CAR possesses a singlet diradical ground state and several singlet-triplet excitation processes within 2 kcal mol⁻¹ in the solid-state by heating. This variety of magnetic transitions are available by the intramolecular conformational mobility of the central azobenzene regarding the terminal "anchored" phenoxyl sites. At low temperature, heat absorption provokes the preferential excitation of low energy torsional vibrational modes of the "freer" azobenzene driving the π-conjugated structure towards a more aromatic form, or "normal" quinoidal→aromatic transformation. Then, this aromatic structure steps back to a less-diradical structure thanks to the additional thermal and intermolecularly cooperative population of a molecular bicycle pedal vibrational motion that conducts heat absorption and produces a thermalized molecular structure with smaller diradical character, or "reversed" quinoidal→aromatic change. In consequence, we describe the cohabitation of "normal" and "reversed" quinoidal→aromatic transformations in a π-conjugated molecule accompanied by the modulation of the spin density distribution and spin mobility as a function of the temperature in the solid by torsional conformational mobility. We provide a detailed spin-vibration coupling effect featuring a vibronic mechanism where selected torsional vibrations mix the ground and excited states at the origin of the observed electronic and spin changes.

## Results and discussion

**Thermal evolution of bond distances and dihedral angles.** We assess the details of the synthesis and characterization of CARH (non-diradical fully aromatic CAR analog) and CAR in the Supplementary material section (Supplementary Figs. 1–7). We grow single crystals of CARH and CAR by using a slow solvent evaporation method in dichloromethane solution under N₂ atmosphere. We studied the solid-state structure of CAR at several temperatures with main focus in those at 130, 290, and 340 K. The average R factor of these structures is ~6.5%, which is reliable for the forthcoming discussion. We disclose selected bond lengths from the solid-state crystals for CARH and CAR in Table 1 and Fig. 2 (Supplementary Tables 1–6).

We now highlight the following structural features. At 130 K, in Fig. 2 and Table 1: (i) the two phenoxyl substituents in CARH are twisted regarding the central azobenzene core with a dihedral angle of 38.8(2)°, whereas this angle is 5.8(5)° in CAR; (ii) the carbon–oxygen bond length in CAR (1.245(4) Å) is shorter than that in CARH (1.377(1) Å), which is very close to a typical carbon–oxygen double bond, such as that of *p*-terphenoquinone (1.231 Å)[18]; (iii) a significant bond length alternation of $r_2$, $r_3$, $r_4$ in the two phenoxyl benzenoid rings; and (iv) a crucial change of the nitrogen-nitrogen bond $r_{10}$ length from 1.252(2) Å in CARH to 1.300(4) Å in CAR, indicating its partial nitrogen-nitrogen single bond character. All these features indicate a sort of quinoidal-like (hereafter, Q) structure on the phenoxyls of CAR at 130 K.

At 290 K: (i) the single-crystal structure reveals that the nitrogen-nitrogen bond $r_{10}$ (1.257(4) Å) becomes much closer to that of CARH (1.252(2) Å); (ii) the bond length alternations of $r_6$, $r_7$, $r_8$ in the internal benzene rings become smaller; (iii) the carbon–carbon bond $r_5$ (1.478(3) Å) in CAR is almost identical to that of CARH (1.484(1) Å); and (iv) the carbon–oxygen $r_1$ and carbon–carbon bonds $r_2$, $r_3$, $r_4$ do not show significant changes at 290 K. These points suggest an exclusive transformation of the azobenzene core at 290 K attaining a structure closer to that typical of aromatic azobenzene. Overall, the bond length changes from 130 to 290 K are mainly in the azobenzene core documenting a decreasing quinoidal character in favor of a more aromatic-like shape (hereafter, A). This type of evolution is typically found in diradicals and can be termed as a "normal" quinoidal→aromatic transformation.

The most significant changes at 340 K compared with those at 290 K are: (i) the length of $r_5$ is 1.449(3) Å shorter than that at

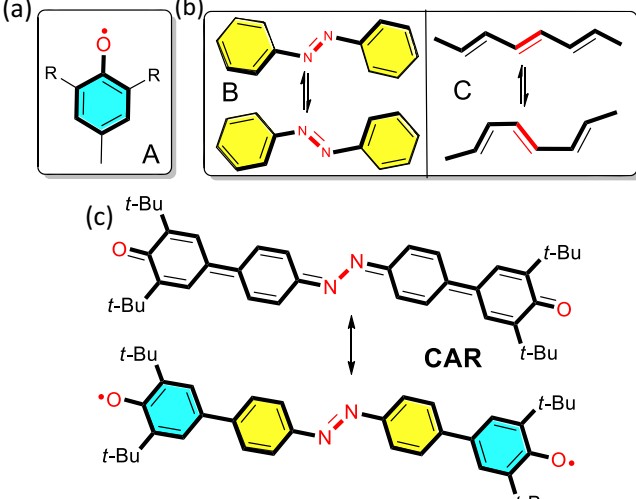

**Fig. 1 Phenoxyl and azobenzene building blocks of CAR and the bicycle pedal motion.** Phenoxyl (**a**) and azobenzene (**b**) building blocks of CAR with the bicycle pedal motion in azobenzene and in oligoenes. **c** Extreme quinoidal/aromatic canonical forms of CAR (note the differences between these extreme forms and the quinoidal-like, aromatic-like, and pseudoquinoidal forms discussed below that always correspond to hybrids with sizeable weights of the two extreme forms).

**Table 1 Selected bond lengths (Å) and dihedral angles at different temperatures of CARH and CAR.**

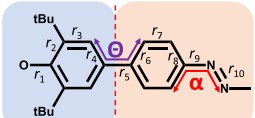

| Bond (Å) | CARH | CAR | | | | |
|---|---|---|---|---|---|---|
| | — | 130 K | 200 K | 250 K | 290 K | 340 K |
| $r_1$ | 1.377(1) | 1.245(4) | 1.240(2) | 1.239(3) | 1.242(4) | 1.238(4) |
| $r_2$ | 1.413(2) | 1.477(5) | 1.477(3) | 1.481(3) | 1.475(4) | 1.483(4) |
| $r_3$ | 1.392(2) | 1.357(4) | 1.356(2) | 1.351(3) | 1.358(4) | 1.358(4) |
| $r_4$ | 1.389(2) | 1.428(5) | 1.420(2) | 1.421(3) | 1.416(4) | 1.414(4) |
| $r_5$ | 1.484(1) | 1.433(4) | 1.436(2) | 1.440(3) | 1.478(3) | 1.449(3) |
| $r_6$ | 1.398(2) | 1.417(5) | 1.429(2) | 1.422(3) | 1.390(3) | 1.419(4) |
| $r_7$ | 1.388(2) | 1.364(4) | 1.362(3) | 1.358(4) | 1.390(3) | 1.366(4) |
| $r_8$ | 1.387(2) | 1.403(5) | 1.395(3) | 1.393(4) | 1.390(3) | 1.379(5) |
| $r_9$ | 1.424(1) | 1.393(4) | 1.391(2) | 1.400(3) | 1.430(3) | 1.426(4) |
| $r_{10}$ | 1.252(2) | 1.300(4) | 1.288(3) | 1.285(3) | 1.257(4) | 1.253(4) |
| **Dihedral (°)** | **CARH** | **CAR** | | | | |
| $\Theta$ | 8.76 | 5.78 | 6.81 | 7.38 | 8.96 | 8.36 |
| $\alpha$ | 11.54 | −0.87 | −0.39 | 0.21 | 1.15 | 0.01 |

The dihedral angles $\Theta$ and $\alpha$ are those involved in the twisting of phenoxyl rings and in the bicycle pedal motion of the central azo group of CAR.

290 K (similar to 130 K); and (ii) the $r_7$ CC bond length decreases by 0.024 Å becoming similar again to that at 130 K. These 290→340 K changes indicate the regaining of partial quinoid character at 340 K or pseudoquinoidal structure (hereafter, PQ). In diradicals and in π-conjugated molecules, one always observes a continuum in the "normal" quinoidal→aromatic transformation on heating what makes certainly unexpected this structural A→PQ inversion or "reversed" quinoidal→aromatic transformation we discover in CAR.

The thermal variation of the dihedral angles in Fig. 2 (Supplementary Figs. 4 and 5) uncovers two distinctive effects. First, an inter-ring torsional motion as the temperature increases to 290 K in which dihedral angles between the connected phenoxyl and azobenzene rings vary from 5.8(5)° at 130 K to 9.0(3)° at 290 K, and then a reduction up to 8.4(4)° passing to 340 K. Second, the two-terminal phenoxyls are twisted in opposite ways with respect to the central azobenzene thereby assuming the same absolute angle value but with opposite signs. The simultaneous presence of the same terminal group twisted in opposite directions in the same molecule and crystal environment, at the same time associated with the opposite orientation of the N=N group, suggests the active role of the bicycle pedal motion well known in crystals of azobenzene derivatives[16]. This dihedral angle evolution delineates a combined effect of the pedal motion jointly acting with the phenoxyl twist owing to the existence of rotational mobility in the azobenzene core, which is expected to increase with temperature. In line with this, we have two shreds of evidence: (i) the crystal structure shows that the CCNN dihedral α, associated with the bicycle pedal motion of the two CC adjacent to the central NN, changes sign with temperature: although the value is small, the sign change further corroborates the rotational mobility; and (ii) the crystal structure disorder found at 340 K in the azobenzene group owing to the bicycle pedal motion (Supplementary Fig. 4).

*Physical properties.* We measured the magnetic susceptibility of solid CAR using a Quantum Design MPMS 5XL SQUID magnetometer and a value of −0.000295 emu mol$^{-1}$ was used as Pascal's correction[19] for diamagnetism of the sample and holder (Supplementary Fig. 8). The plot of χT-T in Fig. 3a shows a curvilinear relationship, Furthermore, the χT value (0.32 emu K mol$^{-1}$) of CAR at 300 K is smaller than the theoretical value of an ideal diradical (0.75 emu K mol$^{-1}$) indicating antiferromagnetic coupling in the ground electronic state. By fitting the curves with a modified Bleaney–Bowers equation[20], the single-triplet energy gap ($\Delta E_{ST}$) amounted to −3.68 kcal mol$^{-1}$.

The ESR spectra (Fig. 3b) showed characteristic triplet patterns at 130 K arising from thermally populated triplets. These are (i) ESR resonances are split by the zero-field splitting; and (ii) we observed the half-field forbidden transition ($\Delta m_s = \pm 2$) as the characteristic signal of the triplet at 168 mT at 130 K (Fig. 3a inset), which is rarely found in thermally excited triplets of organic singlet diradicals[21]. The zero-field splitting parameters amounted to $D = 0.00503$ cm$^{-1}$ and $E = 0.00013$ cm$^{-1}$ from the spectral simulation. From the $D$ value, we calculated the average distance between the two triplet unpaired electrons ($D \sim 1/d^3$) as $d = 8.04$ Å, which is shorter than the length of azobenzene moiety (9.094 Å), suggesting that the spin density is mostly in the molecular center at 130 K in agreement with the remaining phenyls attaining a more quinoidal shape (Q form). We also performed the variable temperature ESR measurements (Supplementary Fig. 9) and estimated the $\Delta E_{ST}$ value to be −3.41 kcal mol$^{-1}$ by fitting the curves with the Bleaney–Bowers equation (Supplementary Fig. 10) from which we obtained a radical impurity amounting to 1.7%. The ESR fitting agrees well with the result from the SQUID.

The solid-state optical absorption spectrum of CAR in Fig. 3c (Supplementary Fig. 11) shows two weak bands at 888 and 997 nm due to the S$_0$→S$_1$ transition to a dark double exciton state typically existing in diradicals[22]. The strongest band at 645 nm owns to the S$_0$→S$_2$ excitation. Figure 3d represents the Raman spectra of CAR taken with the Raman excitation laser lines at 633 nm (in resonance with the 645 nm electronic absorption) and at 1064 nm (in pre-resonance Raman conditions with the 997 nm band). The two spectra share the features at 1597 and 1563 cm$^{-1}$ (blue shaded 1 and 2 bands in Fig. 3d) while the 1064 nm spectrum has additional bands at 1540–1530 cm$^{-1}$ (red shaded 3 bands in Fig. 3d), which vibrationally identifies the double exciton state of the CAR diradical.

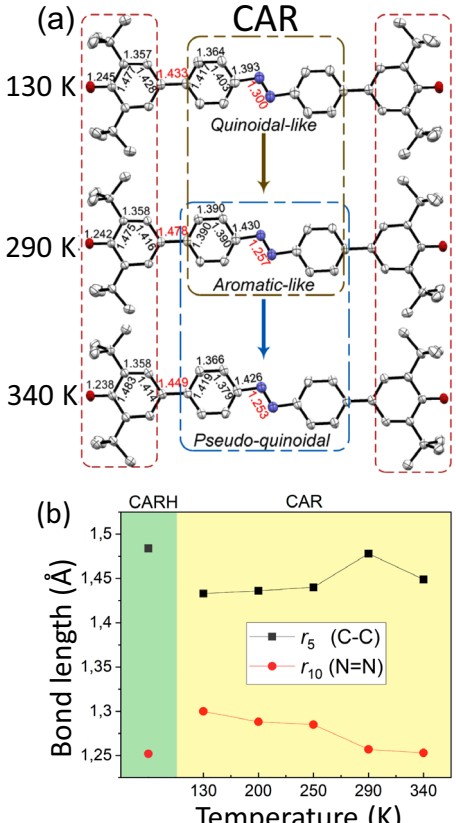

**Fig. 2 Temperature-dependent single-crystal structures of CAR. a** Single-crystal X-ray structure with selected bond lengths of CAR at 130, 290, and 340 K with thermal ellipsoids at the 50% probability level. H atoms are not shown for clarity. **b** Selected bond lengths (Å) values of CARH and CAR.

We further obtained the Raman spectra with the 633 and 785 nm laser excitations as a function of the temperature in Fig. 4, which shows the Raman spectra at 130, 290, and 340 K. Two intervals are relevant: (i) at high wavenumbers, where two main Raman bands ~1600 and 1560 cm$^{-1}$ appear due to the CC bonds stretches of the azobenzene rings [i.e., $\nu(CC)_{ph}$] and to the CC bonds stretches of the phenoxyl rings [i.e., $\nu(CC)_{phO}$] respectively; and (ii) at medium wavenumbers ~1150 cm$^{-1}$ where two bands are detected related with the CN bond stretchings of the azo group connecting to the benzenes rings [i.e., $\nu(CN)_{azo}$] (see electronic supporting information for the theoretical vibrational Raman spectrum–Supplementary Fig. 12 and the vibrational assignment in terms of normal modes in Supplementary Fig. 13).

From the relative intensity of these pairs of bands in each interval, we can assess the thermal evolution of the molecular structures. In the high wavenumber section of the 785 nm Raman spectrum at 130 K, the intensities of the 1600 [$\nu(CC)_{ph}$] and 1560 cm$^{-1}$ [$\nu(CC)_{phO}$] bands (A1/A2 in Fig. 4e) are similar in line with similar quinoidal contributions in the two types of benzene rings at this temperature. In the 290 K Raman spectrum, the $\nu(CC)_{phO}$ band clearly decreases regarding the vicinal $\nu(CC)_{ph}$ one, in agreement with gaining of aromatic character in the azobenzene rings. In the 340 K spectrum, however, the $\nu(CC)_{phO}$ band regains intensity relative to its parent in accordance with a partial recovery of quinoidal character in the azobenzene rings. We found the same behavior in another experiment with the 633 nm laser excitation also in Fig. 4. In the medium wavenumber region of the 785 nm Raman spectra, a similar discussion can be done with the relative intensities of the two $\nu(CN)_{azo}$ bands (i.e., denoted as $\nu(CN)_{azo1}$ and $\nu(CN)_{azo2}$); hence, heating from 130 to 290 K produces a

redistribution of the intensity consisting in a decrease of the $\nu(CN)_{azo2}$ band, a trend that reversed by further heating to 340 K. These findings support a sequence of quinoidal→aromatic transformation on 130→290 K followed by an aromatic→pseudo-quinoidal post-conversion at 340 K. For the experiment with the 633 nm laser excitation in the medium wavenumber region, the thermal evolution of the Raman spectra, though less clearly, also shows the same behavior. Furthermore, the theoretical Raman spectra calculated on the experimental geometries at 130, 290, and 340 K (Supplementary Fig. 14) show changes in nice agreement with the experimental observations. In particular, in Supplementary Fig. 14, we compared the theoretical Raman spectrum obtained in the XRD experimental geometries at 290 K of CAR with its FT-Raman spectrum taken with the 1064 nm laser excitation. A good fitting exists between the experimental and theoretical vibrational spectra supporting the suitability of the structural discussion based on XRD geometries despite the perturbative affectation of disorder at high temperatures.

*Singlet-triplet gaps ($\Delta E_{ST}$) and thermal spin distribution.* To further understand the magnetic structure of CAR, we carried out density functional theory (DFT) calculations at B3LYP/6-31G(d) and M06-2X/6-311G(d) levels (Supplementary Tables 7 and 8)[23–27]. Considering CAR as an isolated entity in the vacuum, calculations predict an open-shell singlet ground state and $\Delta E_{ST}$ gaps at different levels of theory of −0.43 kcal mol$^{-1}$ (M06-2X/6-311G(d) and −0.94 kcal mol$^{-1}$ (UB3LYP/6-31G(d)). On the x-ray molecular structures at 130, 290, and 340 K, UB3LYP/6-31G(d) quantum chemical calculations predict three different $\Delta E_{ST}$ values of −4.94, −1.58, and −2.34 kcal mol$^{-1}$, respectively, whose average value, −2.95 kcal mol$^{-1}$, is in very good agreement with the current experimental value of −3.68 kcal mol$^{-1}$ from SQUID measurements. Furthermore, calculations estimate a diradical character value of $y_0 = 0.93$ for the in-vacuum broken-symmetry (BS) optimized structure (M06-2X/6-311G(d) level), whereas we calculated values of $y_0 = 0.81$, $y_0 = 0.92$, and $y_0 = 0.88$ for the crystal structures of CAR at 130, 290, and 340 K, respectively. This theoretical description indicates the modulation and tuning of the diradical character of CAR in its thermal forms referred to as Q form ($y_0 = 0.81$), A form ($y_0 = 0.92$), and PQ form ($y_0 = 0.88$). Hence, CAR has three singlet-triplet excitation processes due to the progressive thermal conversion of its structure. Unfortunately, these three $\Delta E_{ST}$ gaps could not be resolved experimentally by SQUID magnetometry, which overall detects one single process.

We calculated the spin distributions in the crystal structures at different temperatures at the M06-2X/6-311G(d) level in Fig. 5a (Supplementary Fig. 15 and Table 9), which shows selected spin population values of specific atoms. As the temperature increases from 130 to 290 K, the spin density decreases in the azobenzene core (Fig. 5b) and moderately increases in the phenoxyls. The most obvious change of spin density resides in the N atom which decreases from 0.174 to 0.081 from 130 to 290 K at the same time that the spin density of the azobenzene core becomes very small. At 340 K, the integrated spin density of the N atoms and of the azobenzene unexpectedly increases. Correlating spin distribution and molecular structures, at 130 K the Q form spreads out the spin density uniformly over the whole skeleton of CAR (light blue circles and red lines in Fig. 5). The transformation into the A form at 290 K largely concentrates the spin density in the terminal phenoxyls (red circles and black lines in Fig. 5). Finally, at 340 K, the PQ structure again redistributes the spin density towards the central moiety.

## Mechanism of spin delocalization

*Transformations among different spin-delocalized structures.* The thermal transformation of CAR along the $y_0$ coordinate

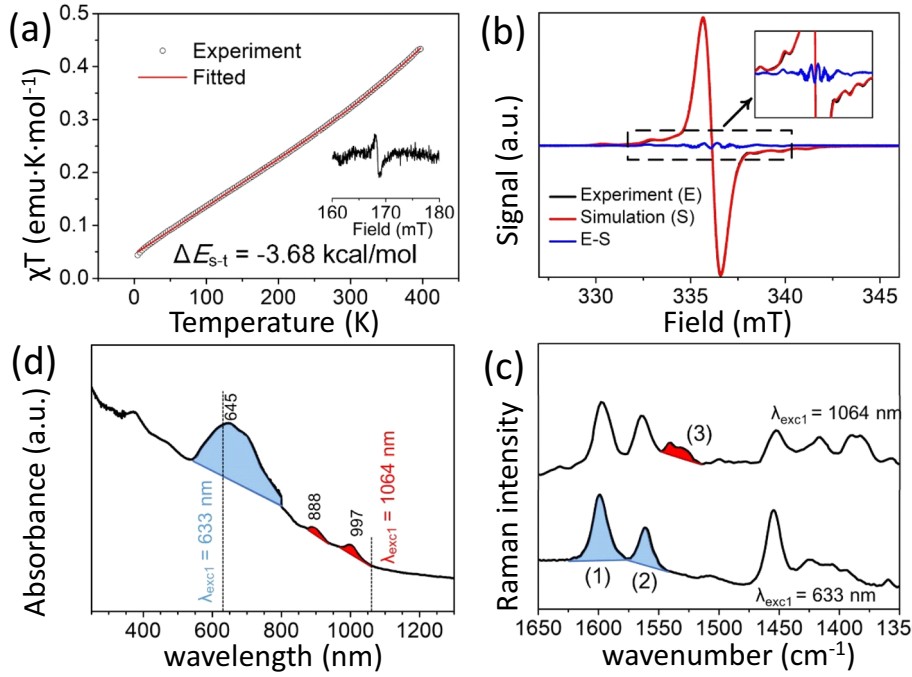

**Fig. 3 Solid-state magnetic and spectroscopic properties of CAR. a** Temperature-dependent plots of χT versus T and fitted χT–T curve (solid line) for CAR measured at 1.0 T from 2 to 400 K. The half-field transitions (Δ$m_s$ = ±2) are shown as insets. **b** ESR spectra of 1 mmol L$^{-1}$ CAR in a frozen glass (toluene) at 130 K. **c** UV-Vis-NIR electronic absorption spectrum in solid-state of CAR at room temperature; **d** room temperature solid-state Raman spectra of CAR taken with the 633 nm and 1064 nm excitation laser lines.

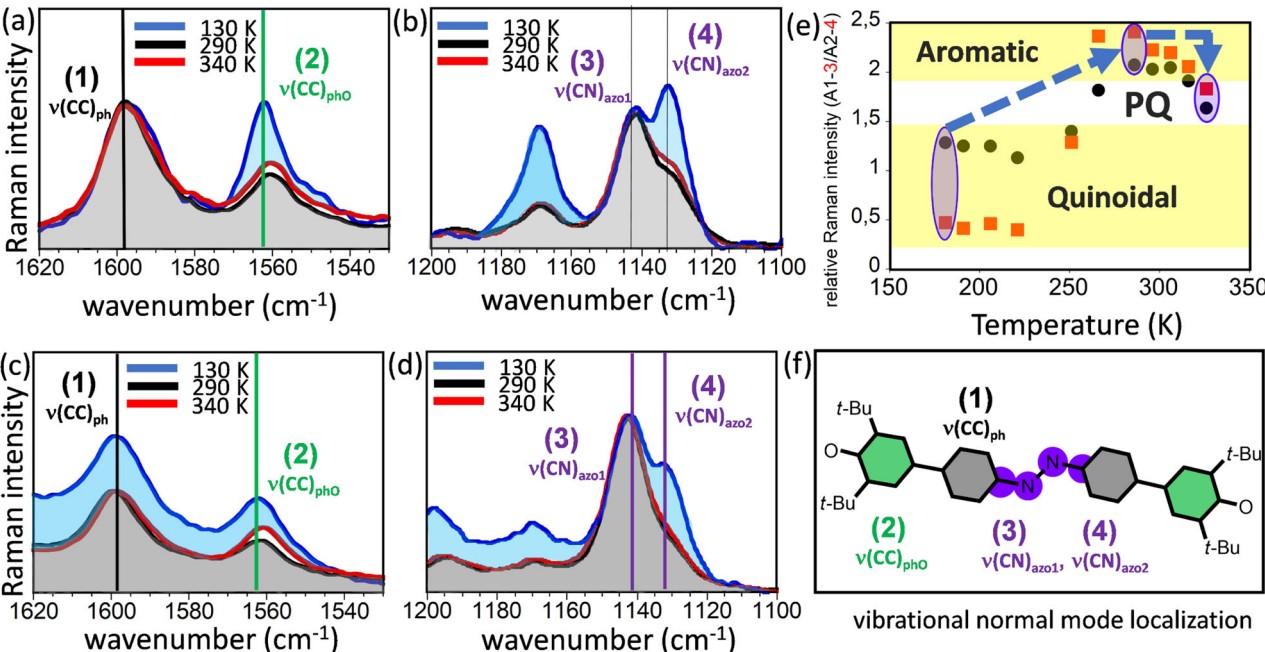

**Fig. 4 Variable temperature vibrational Raman spectroscopy.** Solid-state variable temperature Raman spectra of CAR taken with the 785 nm (**a**) and (**b**) and with 633 nm excitation wavelength (**c**) and (**d**) in two different vibrational intervals where the four important (strongest) bands are denoted as (1) ν(CC)$_{ph}$; (2) ν(CC)$_{phO}$; (3) ν(CN)$_{azo1}$; and (4) ν(CN)$_{azo2}$. **e** Representation as a function of the temperature of the ratio of intensities of 1 (A1) and 2 (A2) as black circles and 3 (A3) and 4 (A4) bands as red squares as spectroscopic markers of the degree of quinoidal/aromatic character, which delineates three regions: small ratio means more quinoidal, large ratio more aromatic and pseudoquinoidal (PQ) is in between. **f** Topologies of the vibrational modes from the theoretical Raman spectra in Supplementary Figs. 12–14.

corresponds to two separated processes. An (0.81→0.92) increase for Q→A followed by a (0.92→0.88) decrease for A→PQ, where the latter step cannot be viewed as a way back the evolution of the former. This is deduced from calculations of the Huang-Rhys

factors[28] alongside the two above structural transitions (Supplementary Table 10), which reveal that the set of vibrational normal mode contributions that conduct the two transformations are different (e.g., there is not a vibrational reaction coordinate that

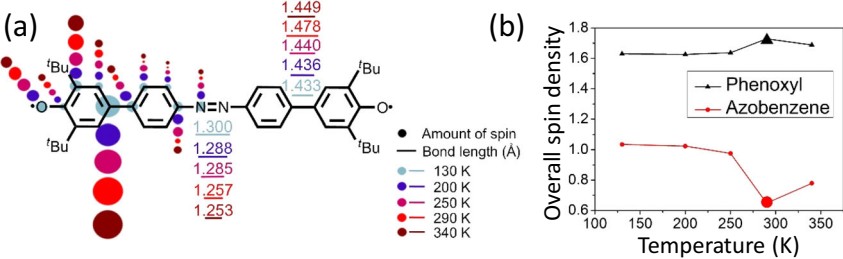

**Fig. 5 Spin density distribution of CAR. a** Selected bond lengths and calculated spin population of CAR at different temperatures. **b** The integral spin density changes of phenoxyl and azobenzene moieties at temperatures of 130, 200, 250, 290, and 340 K.

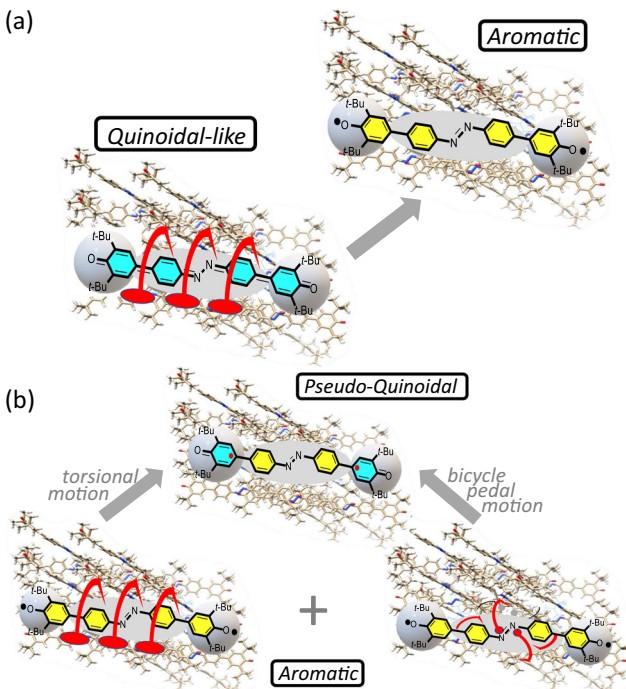

**Fig. 6 Sequential vs simultaneous action of torsion and bicycle pedal motions.** Simulations (not at scale) of the dihedral torsion motion of the azobenzene relative to the phenoxyls provoking the Q→A transformation (**a**), and of the combined torsional plus bicycle pedal motions of the central azobenzene moiety provoking the A→PQ transformation (**b**).

sequentially transforms among them). Hence, the portion of the potential energy surface (PES) that drives the structural changes of CAR may be figured out in a multidimensional space where stretching coordinates are not only involved but additional torsional nuclear displacements also occur that are highly affected by the solid-state environment. The multidimensional PES of CAR agrees with the simultaneous changes observed experimentally for the π-bond distances and for the dihedral angles from x-ray data.

*Crystal effects in the spin transformation: QM/MM calculations.* Given the distinctive role of torsional motion, we explored the PES of CAR with QM/MM calculations (Supplementary Figs. 16–19) as a function of (i) the inter-ring torsion around the CC bonds connecting the azobenzene moiety with the phenoxyl terminal units (Fig. 6a), and (ii) the bicycle pedal motion involving the two CC bonds adjacent to the central azo group (Fig. 6b). The three crystal geometries at 130, 290, and 340 K were considered inside a cluster formed by 18 molecules completely surrounding the central one, described at QM level, to include the effects (via MM potential) of the nearby molecules in the crystal.

The simultaneous torsion around the two CC bonds may occur by twisting the whole azobenzene moiety inside the crystal cavity with respect to the terminal phenoxyls, which remain almost positionally cleaved (dumbbell structure), such as drawn schematically in Fig. 6. Similarly, also the bicycle pedal motion in Fig. 6 can occur by leaving the terminal moieties of the CAR molecule almost fixed. We thus realize that these are key motions that jointly provide with torsional solid-state mobility to azobenzene.

A significant point is that the QM/MM computed PES along with the two Q→A and A→PQ selected torsional paths are asymmetric (Fig. 7 and Supplementary Figs. 17–19) with respect to the planar structure in contrast with the isolated/in-vacuum computed PES, which is symmetric with two degenerate energy minima separated by a small barrier (i.e., such as in biphenyl)[29]. The asymmetry is more marked for the torsion of the whole azobenzene with respect to the phenoxyl groups (Fig. 7) compared with the bicycle pedal motion as determined by the interaction with neighbor molecules in the crystal, otherwise kept fixed while scanning the PES. The PESs alongside the 130–290 K transformation do not intercept each other by varying the torsional angle (Supplementary Fig. 17) indicating that the Q→A transformation is progressive and might be driven by entropy effects based on the increase of molar entropy on going towards a more distorted structure. The increase of entropy is conducted by thermal excitation/population of low energy torsional modes which is not reflected in the PESs behavior.

The consideration of the 290 K and 340 K PES's shows a crossing around 30° in which the A form is favored for θ > 30° and the PQ form is preferred for θ < 30°. This crossing could be favored and modulated by coupling with the bicycle pedal motion. In fact, if we look at a CAR molecule inside the crystal with a focus on the two closest CAR molecules (above and below the central one in Supplementary Fig. 5), the θ angle is found positive or negative only when coupled with a specific orientation of the azo group. This suggests that the torsion of the azobenzene may be coupled with the bicycle pedal motion. Therefore, it is the joint action of the torsion of azobenzene and of the bicycle pedal motion that assists the crossing between PESs and drives the geometry change occurring with the temperature increase, such as schematized in Fig. 6. The A→PQ transition takes place as a way to reduce the interaction energy in a given interval of dihedral angles.

Taking into account the distinctive out-of-plane topologies of the twisting and bicycle pedal motions, we rationalized the observed transformations in the following way: (i) low energy torsions exclusively drive the "normal" Q→A change by which always favors the aromatic structures[30]; however, (ii) when exciting the higher energy bicycle pedal vibrational states together with the torsional vibrations, the system prefers a less-distorted thermalized molecular structure, PQ, with smaller diradical character, outlining a "reversed" quinoidal→aromatic transformation.

In conclusion, we successfully synthesized and characterized a CAR, which possesses a singlet open-shell ground electronic state.

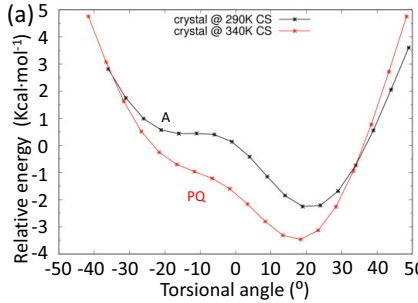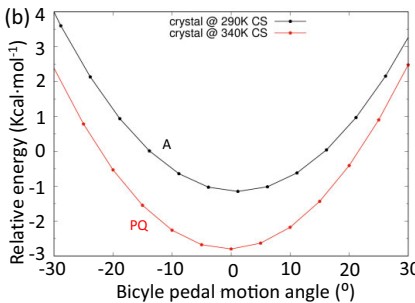

**Fig. 7 Potential energy surface dependence of CAR with torsion and bicycle pedal motions. a** Torsional angle potential energy profiles of the CAR molecule inside the crystal structure at 290 K (A form) and of that at 340 K (PQ form). **b** Potential energy profiles of the CAR molecule inside the crystal structure at 290 K (A form) and of that at 340 K (PQ form) alongside the bicycle pedal motion distortion. Calculations were carried out with the QM/MM ONIOM model on the cluster of CAR molecules, B3LYP/6-31G* level with electronic embedding for the central molecule, Dreding force field, and charges determined with the Qeq approach for the MM part.

CAR has a singular solid-state magnetic structure characterized by the thermal tuning of the spin distribution and of singlet-triplet gaps as a result of a unique solid-state structural modulation due to the intramolecular mobility/flexibility imprinted by the dumbbell structure of the azobenzene moiety. These magnetic processes involve three structures, all with large diradical character: a quinoidal-like form ($y_0 = 0.81$) at low temperature, an aromatic-like form ($y_0 = 0.92$) at intermediate values and a pseudoquinoidal-like form ($y_0 = 0.88$) at high temperatures. These transitions are progressively driven by the population of torsional intramolecular vibrational modes of the azobenzene group. Distinctively, whereas heating at low temperatures produces a "normal" Q→A step, the case of the "reversed" quinoidal→aromatic transformation or aromatic→pseudoquinoidal (A→PQ) transformation occurs conducted by a torsional/bicycle pedal vibrational mode favoring a more planar pseudoquinoidal form. This invokes a vibronic coupling mechanism where selected vibrational normal modes mix the ground electronic state with excited states, resulting in electronic structures with added bond covalency (i.e., more quinoidal character) that reduce the total molecular energy. These two vibronic processes highlight mechanisms of single (torsions) and double (torsions plus bicycle pedal) electron-vibration couplings, which turn out in such unprecedented thermal spin mobility in the CAR diradical.

## Methods

**Synthesis**. CARH was readily synthesized using 4,4'-dibromoazobenzene and 2,6-di-*tert*-butylphenoxyl substituents under a typical Suzuki-Miyaura cross-coupling reaction condition with Pd(PPh$_3$)$_4$ as catalyst and Na$_2$CO$_3$ as a base in THF solution. Subsequently, when treated with excess lead (IV) oxide in dichloromethane CARH was easily oxidized to CAR in high yield (see supporting information).

**Raman spectroscopy**. The Raman spectra were recorded by using the 633 and 785 nm excitations of a Bruker Senterra Raman microscope by averaging spectra during 50 min with a resolution of 3–5 cm$^{-1}$. A CCD camera operating at −50°C was used. The spectra were collected using the 1 × 1 camera of the mentioned microscope. Variable temperature Raman spectra were obtained in a Linkam cell working up to 80 K incorporated in the microscope camera of the Senterra spectrometer.

**DFT calculations**. All calculations were performed with the Gaussian 16 program suite. For isolated CAR molecules, we used DFT with M06-2X exchange-correlational functionals and employed the 6-311G(d) basis set for all atoms. Additional calculations, including Huang-Rhys parameters, were carried out using B3LYP functionals and the 6-31G* basis set. Full geometry optimizations were carried out at the (U)M06-2X/6-311G(d), B3LYP/6-31G* levels, and the obtained stationary points were characterized by frequency calculations. The spin correction for the

singlet-triplet energy gap was calculated using Yamaguchi's procedure which includes the energy difference between BS singlet and open-shell triplet and the correction for spin contamination.

## Data availability

Data supporting the findings of this study are available within the article and its Supplementary Information files. Crystallographic data generated in this study for CARH and CAR have been deposited in the Cambridge Crystallographic Data Center (www.ccdc.cam.ac.uk) under accession codes CCDC-2006244(CARH), CCDC-2006243(CAR 130K), CCDC-2006239(CAR 200K), CCDC-2006240(CAR 250K), CCDC-2006241(CAR 290K) and CCDC-2006242(CAR 340K). The authors declare that all other data supporting the findings of this study are available within the paper [and its supplementary information files]. Other spectroscopic Raman and theoretical data are available on request from the authors. Data for this research are presented in https://www.researchsquare.com/article/rs-116016/v1 (https://doi.org/10.21203/rs.3.rs-116016/v1).

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

## Acknowledgements

The authors acknowledge financial support from the National Natural Science Foundation of China (61805034) and the Scientific Research Foundation of UESTC for Young Teachers (Y03019023601008007). We thank MINECO/FEDER of the Spanish Government (PGC2018-098533-B-I00), Junta de Andalucía (UMA18FEDERJA057) and Research Central Services (SCAI) of the University of Málaga. F.N. and Y.D. gratefully acknowledge the financial support from the University of Bologna (RFO) and computational resources from CINECA through an ISCRA (Italian Super Computing Resource Allocation) C project. Y.D. acknowledges MIUR for her Ph.D. fellowship.

## Author contributions

Y.S. conceived the project, and designed and carried out the experiments. F.M. analyzed the data and wrote the manuscript. Y.H.Z. conceived the project and played a critical role in discussions of the experimental design, project direction, experiments and results, and preparation of the manuscript. H.J.C. carried out and analyzed the SQUID and ESR magnetic measurements. G.D.X. calculated and analyzed the spin density distribution. S.M.Q. and J.C. performed the Raman spectroscopic measurements. Y.D. and F.N. performed the calculations and analyzed the data from these computations. J.C. and F.N. provided the mechanism of spin-vibrational coupling and discussed it in the context of the experimental results. All authors discussed the results and commented on the manuscript.

## Competing interests

The authors declare no competing interests.
