## [Peer Review File · Nature Communications]

Reviewers' Comments:

Reviewer #1:

Remarks to the Author:

This manuscript by Shen et. al describes an interesting study on the diradical character of an azobenzene molecule as a function of differing temperatures. The authors attribute an unusual bicycle pedal motion of the molecule to populating a more pseudo-quinoidal structure at elevated temperature (340 K). This area is interesting, and the proposed structural driven change in electronic structure is novel. The major weakness of the paper, in my opinion, is that the only experimental data that the authors have that supports this conclusion is a crystal structure at 340 K. While this data is consistent with their hypotheses, crystal structures at this temperature will have some limited accuracy due to vibrational motions etc. My recommendation is that this paper be reconsidered after revisions to provide additional experimental support for the proposed structural change.

Specific points are below:

1. My major concern is that the only experimental data that supports this change is provided by a single crystal structure. The XRD data is generally good, with some exceptions as noted below, but this is still only a single piece of experimental evidence for their conclusions. The magnetic data does not show any changes which would be consistent with the proposal. What other data can the authors acquire that would support this? Can they get variable temperature spectroscopic or vibrational data to support this change? I think that the changes are subtle enough that some corroborating data would strengthen their arguments.
2. All discussions of bonding in the text and tables should have ESD's associated with the bonds. Similarly, angles need to have ESD's provided.
3. Measured physical constants should also have ESD's reported.
4. Some additional information on the refinement of the crystal data would be valuable. The ellipsoids look a little strange, what restraints were applied? If large restraints were applied to make the ellipsoids more isotropic they should be relaxed, or the authors should at least compare the changes in bond lengths with or without the restraints. This is particularly crucial given the importance of the structural data in the author's arguments.
5. The authors invoke a pedal motion to explain the different structures (as cited in reference 13). However, as stated in ref. 13, "disorder in crystal structures is an important indicator of pedal motion in the crystal". Why do the authors not see any evidence of this disorder? If a pedal motion were present, wouldn't the atom positions be disordered? It is possible that only one conformer would be favored, as discussed in the cited reference, but the author's structural data seems to suggest that there are two conformers close in energy. I might expect that some evidence of disorder would appear at 340 K. At the very least, this needs to be explained more clearly.
6. What is the feature at 55 K in the magnetic data?
7. It seems that some more recent citations to aggregation or thermally modulated diradical character are absent. Examples include: Chem. Sci. 2016, 7 (10), 6514–6518. and 2019, 10 (46), 10733–10739., as well as J. Am. Chem. Soc. 2018, 140, 43, 14308–14313 and 2020 142 (41), 17670-17680.

Reviewer #2:

Remarks to the Author:

In this manuscript, the authors reported an azobenzene-based organic diradical with a dumbbell shape. The correlation between its solid-state flexibility and spin diradical structure has been illustrated. As shown in the X-ray temperature-dependent studies, the authors found that the spin distribution and the low-to-high spin magnetic transitions can be modulated, quinoidal to aromatic transformation at low temperatures can be driven by the intramolecular rotational motions, and a

"reversed" aromatic to quinoidal change at high temperatures can be activated by a bicycle pedal motion. A unique and spin-vibration mechanism has been revealed and clearly discussed. This work is of general interests for chemists majoring in diradical chemistry and of high value in organic spintronics. Besides, the paper is well organized with a high-level scholar presentation. Before its consideration for formal publication, I would suggest some revisions to be made.

1. Although the rich transformations between aromatic and quinoidal canonical forms can be clearly observed in the single crystal, the magnetic susceptibility measurements of CAR should also be performed in the crystal state. It is known that there may exist several phases in multicrystalline or powder states that may affect the discussed transformations. So please indicate this information in the paper.

2. Following the first suggest, the other measurements should also be performed under the crystal state. If it is not the case or it is difficult to perform such measurements, I would suggest the authors check XRD of the solid and confirm the molecular stacking is consistent in crystals and powders.

3. Since the triplet state is thermally activable, we should keep in mind that the triplet state and singlet state coexist in the crystal and the geometry information actually is a statistical data. In other words, with the increasing temperature, the triplet state will populate, which may affect the molecular geometry and maybe the packing. The authors are suggested to carefully consider this possible issue.

4. A clear explanation on the varied intramolecular torsional motion under different temperatures should be provided in the abstract, introduction, and conclusion parts, which may induce more consideration of this design and get a larger influence.

5. About the importance of such a unique phenomenon, the authors are suggested to put more efforts. I think it is not just a new phenomenon, and my feel is that it should be quite meaningful for organic spintronics considering its tunable spin distribution.

6. There are a few typos in the main text and in the information. Please check and correct them. For example, in "... a solid state building scaffold able to bringing intramolecular torsional mobility...", "bringing" should be corrected; In "... bicycle pedal motion in in azobenzene (B)...," please remove one "in".

Reviewer #3:

Remarks to the Author:

Shen et al reported that they prepared new azobenzene-based organic diradical. The diradical has been proven as an open-shell singlet ground state by EPR, SQUID and DFT calculation. They claimed that the geometry diradical is flexible with temperature, accompanied with spin density changes. However, based on the std (3-4, and higher at high T in cifs) of bond lengths of X-ray structures, it is hard to see there is any change of geometries at different temperatures. The line of XT is very smooth, consistent with the property of a singlet diradical, indicating there is no changes of the electron coupling. Since the std is so large, that DFT (spin density and γ values) based on X-ray structures result is totally unreliable. In words, the finding is just a normal diradical with an open-shell singlet ground state, and basically no alternation of geometries with temperature based on crystal structures. The current version of the manuscript is unpublishable and is supposed to be rewritten. Several points need consideration:

1. ST gap difference between determined values by EPR and SQUID (1.31 vs 3.41 Kcal/mol) is large. One of them must be incorrect.

2. Thermal driven geometry change of organic diradicals have been reported before, especially by Juan and Wang et al. Those literature work should be clearly described in the text.

3. STD of geometry parameters shod be added in the text and tables.

REVIEWER #1:

Reviewer point 1. My major concern is that the only experimental data that supports this change is provided by a single crystal structure. The XRD data is generally good, with some exceptions as noted below, but this is still only a single piece of experimental evidence for their conclusions. The magnetic data does not show any changes which would be consistent with the proposal. What other data can the authors acquire that would support this? Can they get variable temperature spectroscopic or vibrational data to support this change? I think that the changes are subtle enough that some corroborating data would strengthen their arguments.

Authors' response 1. Solid state Raman spectra in the polycrystalline sample of **CAR** have been obtained as a function of the temperature (for completeness two excitation Raman lines have been analyzed). The new data, their description and discussion have been incorporated in the new version of the article in pages 9-11 including a new Figure 4. Moreover, quantum chemical calculations of the Raman spectra on the experimental molecular structures at different temperatures have been obtained which nicely reproduced the experimental Raman spectra which therefore support the new discussion. The Raman spectra nicely confirm the previous arguments in the paper and represent a new piece of evidence (such as the reviewer required) in line with the whole discussion. We very much appreciate the comment of the reviewer.

The new theoretical Raman spectra have been included in the new version of the electronic supporting information (new Supplementary Figures 12-14). The text with the Raman discussion has been incorporated in the revised version of the paper in pages 9-11. This text added in the main body of the article follows below:

Raman spectra with the 633 and 785 nm laser excitations have been measured as a function of the temperature and the spectra at 130, 290 and 340 K are shown in Figure 4. Two intervals are relevant: i) at high wavenumbers, where two main Raman bands around

1600 and 1560 cm^{-1} due to the CC bonds stretches of the azobenzene rings [i.e., $\nu(\text{CC})_{\text{ph}}$] and to the CC bonds stretches of the phenoxy rings [i.e., $\nu(\text{CC})_{\text{phO}}$] respectively, appear; and ii) at medium wavenumbers around 1150 cm^{-1} where two bands of the CN bond stretchings of the azo group connecting to the benzenes rings [i.e., $\nu(\text{CN})_{\text{azo}}$] dominate (see electronic supporting information for the theoretical vibrational Raman spectrum - Supplementary Fig. 12 and the vibrational assignment in terms of normal modes in Supplementary Fig. 13).

Figure 4. Solid-state variable temperature Raman spectra of **CAR** taken with the 785 (top) and 633 nm (bottom) laser excitations in two different vibrational intervals where the four important (strongest) bands are denoted as (1) $\nu(\text{CC})_{\text{ph}}$; (2) $\nu(\text{CC})_{\text{phO}}$; (3) $\nu(\text{CN})_{\text{azo1}}$ and (4) $\nu(\text{CN})_{\text{azo2}}$. Top, right: representation as a function of temperature of the ratio of intensities of 1 (A1) and 2 (A2) bands as a spectroscopic marker of the degree of quinoidal/aromatic character which delineates three regions: small ratio means more quinoidal, large ratio more aromatic and pseudoquinoidal (PQ) is in between. Bottom, right: topologies of the vibrational modes from the theoretical Raman spectra in Supplementary Figs. 12, 13 and 14.

The relative intensity of these pairs of bands in each interval can be linked with the thermal evolution of the molecular structures. In the high wavenumber section of the 785 nm Raman spectrum at 130 K, the intensities of the 1600 [$\nu(\text{CC})_{\text{ph}}$] and 1560 cm^{-1} [$\nu(\text{CC})_{\text{phO}}$] bands (A1/A2 in Figure 4) are similar in line with similar quinoidal

contributions in the two types of benzene rings. In the 290 K Raman spectrum, the $\nu(\text{CC})_{\text{phO}}$ band clearly decreases regarding the vicinal $\nu(\text{CC})_{\text{ph}}$ one, in line with a gaining of aromatic character in the azobenzene rings. In the 340 K spectrum, however, the $\nu(\text{CC})_{\text{phO}}$ band regains intensity relative to its parent one in accordance with a partial recovery of quinoidal character in the azobenzene rings. The same is found in another experiment with the 633 nm laser excitation also in Figure 4. In the medium wavenumber region of the 785 nm Raman spectra, a similar discussion can be done with the relative intensities of the two $\nu(\text{CN})_{\text{azo}}$ bands (i.e., denoted as $\nu(\text{CN})_{\text{azo1}}$ and $\nu(\text{CN})_{\text{azo2}}$); hence, heating from 130 to 290 K produces a redistribution of the intensity consisting in a decrease of the $\nu(\text{CN})_{\text{azo2}}$ band an effect that is reversed by further heating to 340 K in accordance with a sequence of quinoidal \rightarrow aromatic transformation on 130 \rightarrow 240 K, followed by an aromatic \rightarrow pseudo-quinoidal post-conversion at 340K. For the experiment with the 633 nm laser excitation in the medium wavenumber region, the thermal evolution of the Raman spectra, though with less clarity, also shows the same behavior. Furthermore, the theoretical Raman spectra calculated on the experimental geometries at 130, 290 and 340 K have been obtained (Supplementary Fig. 14) which show changes in nice agreement with the experimental observations, in particular in the medium wavenumber region).

Reviewer point 2 and 3. All discussions of bonding in the text and tables should have ESD's associated with the bonds. Similarly, angles need to have ESD's provided. 3. Measured physical constants should also have ESD's reported.

Authors' response 2 and 3. The ESD of the bonds and angles were marked in the text and tables in the revised manuscript as follows:

In page 5 and in line 2, “38.76°” was replaced with “38.8(2)°”, “5.78°” was replaced with “5.8(5)°”.

In page 5 and in line 3, “1.245 Å” was replaced with “1.245(4) Å”, “1.377 Å” was replaced with “1.377(1) Å”.

In page 5 and in line 7, “1.252 Å” was replaced with “1.252(2) Å”, “1.300 Å” was replaced with “1.300(4) Å”.

In page 5 and in line 11, “1.257 Å” was replaced with “1.257(4) Å”, “1.252 Å” was replaced with “1.252(2) Å”.

In page 6 and in line 1, “1.478 Å” was replaced with “1.478(3) Å”, “1.484 Å” was replaced with “1.484(1) Å”.

In page 6 and in line 10, “1.449 Å” was replaced with “1.449(3) Å”.

In page 7 and in line 6, “5.78°” was replaced with “5.8(2)°”, “8.96°” was replaced with “9.0(3)°”.

In page 7 and in line 7, “8.36°” was replaced with “8.4(4)°”.

Reviewer point 4. Some additional information on the refinement of the crystal data would be valuable. The ellipsoids look a little strange, what restraints were applied? If large restraints were applied to make the ellipsoids more isotropic they should be relaxed, or the authors should at least compare the changes in bond lengths with or without the restraints. This is particularly crucial given the importance of the structural data in the author’s arguments.

Authors’ response 4. For the single crystal data at 130 K and 200 K, no additional geometric and ADP restraints were applied in the structure refinement. For the single crystal data at 250 K and 290 K, overall ADP restraints were used during the refinement. Removing them did not affect the structure. For the single crystal at 340 K, the N atom ellipsoid seems strange compared to surrounding atoms, due to some disorder and EDAP constraint was used to make the ADP to look more reasonable. Removing them makes N=N bonds a little shorter. Very nicely the appearance of disorder in the 340K structure is compatible with the activation of the bicycle pedal motion which simultaneously produces the aromatic→pseudoquinoidal electronic change. To expand this discussion, please

see response to point number 5 below for this referee.

This new discussion has been added in the revised version of the article in page 5. In addition, in page 5 of revised electronic supplementary information, the corresponding contents and discussion of the structural disorder were added.

Reviewer point 5. The authors invoke a pedal motion to explain the different structures (as cited in reference 13). However, as stated in ref. 13, “disorder in crystal structures is an important indicator of pedal motion in the crystal”. Why do the authors not see any evidence of this disorder? If a pedal motion were present, wouldn't the atom positions be disordered? It is possible that only one conformer would be favored, as discussed in the cited reference, but the author's structural data seems to suggest that there are two conformers close in energy. I might expect that some evidence of disorder would appear at 340 K. At the very least, this needs to be explained more clearly.

Authors' response 5. As stated in ref. 13, “disorder in crystal structures is an important indicator of pedal motion in the crystal”. For the single crystal data of 130 K and 200 K, the influence of disorder is small. For the single crystal data at 250 K, 290 K and 340 K, the results after disorder operation showed two conformations (Figure R1) with one of them being clearly dominating. With the temperature increase from 250 K to 340 K, the ratio of the two conformer populations is constantly changing: 0.903:0.097 (250 K), 0.845:0.155 (290 K), 0.827:0.173 (340 K). These results indicate that a conformational equilibrium takes place in the crystal of **CAR** at every temperature. Therefore, this temperature dependence of populations of the conformers is a further proof of the existence of the pedal motion in **CAR**. Though the number of temperature data and relative populations is small, these have been fitted to a van't Hoff plot producing an energy gap between the states in equilibrium of 1.26 kcal/mol which corresponds to roughly 440 cm⁻¹. Though this value must be taken with caution given the small number of data in the plot, it already suggests the existence of a low energy vibrational mode around 440 cm⁻¹ that would

correspond to the torsional plus double pedal motion that couples to the electronic structure. However, we did not go in more details due to the weakness of the plot but leaves here this description for the benefit of the reading of the reviewer.

Figure R1. The major and minor conformers of disordered crystal structure of **CAR**.

In page 6 of the revised version of electronic supplementary information, the explanation about the change of the two conformers from 130 to 340 K was added.

Reviewer point 6. What is the feature at 55 K in the magnetic data?

Authors' response 6. The SQUID experiment has been measured again and the new χT and T curve is shown in Figure R2, which indicates that the feature at 55 K for the old spectrum was accidental. In the new version of Figure 3 the new SQUID data have replaced the old one.

Figure R2 Temperature-dependent plots of χT versus T and fitted χT - T curve for **CAR** measured at 1.0 T from 2 to 400 K.

Reviewer point 7. It seems that some more recent citations to aggregation or thermally modulated diradical character are absent. Examples include: Chem. Sci. 2016, 7 (10), 6514–6518. and 2019, 10 (46), 10733–10739., as well as J. Am. Chem. Soc. 2018, 140, 43, 14308–14313 and 2020 142 (41), 17670-17680.

Authors's response 7. We have cited these references in the revised manuscript in page 2 and in line 20.

REVIEWER #2:

Reviewer point 1. Although the rich transformations between aromatic and quinoidal canonical forms can be clearly observed in the single crystal, the magnetic susceptibility measurements of CAR should also be performed in the crystal state. It is known that there may exist several phases in multicrystalline or powder states that may affect the discussed transformations. So please indicate this information in the paper. Following the first suggest, the other measurements should also be performed under the crystal state. If it is not the case or it is difficult to perform such measurements, I would suggest the authors the check XRD of the solid and confirm the molecular stacking is consistent in crystals and powders.

Authors's response 1. Owing to the difficulty of growing single crystals, it is impossible to make magnetic susceptibility measurements, and related measurements, in the single crystal state. The criticism of the reviewer referring to the use of a polycrystalline sample instead of a single crystal, however, can be solved by comparing the powder XRD experiment in the polycrystalline sample in Figure R3 below (i.e., the same sample we used for the magnetic measurements) with that obtained by simulation of a XRD pattern considering the structure of the single crystal of 290 K which is also shown in Figure R3. The comparison is good by which we are confident that the magnetic measurements carried out in the polycrystalline sample are closely correlated with those we would obtain by measurement of the SQUID in the single crystal. This result can indicate that the

molecular stacking is consistent in crystals and powders.

In the new version of the article, we have included this Figure R3 in page 12 of electronic supplementary information as Supplementary Fig.7.

Figure R3. The simulative (black line) powder pattern based on the single crystal of 290 K and experimental (red line) powder pattern of **CAR**.

Reviewer point 2. Since the triplet state is thermally activable, we should keep in mind that the triplet state and single state coexist in the crystal and the geometry information actually is a statistical data. In other words, with the increasing temperature, the triplet state will populate, which may affect the molecular geometry and maybe the packing. The authors are suggested to carefully consider this possible issue.

Authors' response 2. The reviewer would agree with us in the fact that the singlet triplet energy gap over which the transitions take place are certainly small or very small. This means that molecular species (the singlet and the triplet) with very similar formation energies would disclose very similar molecular structures. So, in our case, we could not expect significant (even detectable) differences between the structures of the singlet and of the triplet. What we observe by heating is a transformation of the ground electronic singlet diradical state between forms with different portion of aromatic and quinoidal contribution and,

for each of these, there are associated triplets that would be similar to their singlets. So, in principle, this issue should not be affecting considerably in our case and, therefore, our argumentation holds in the same terms.

Reviewer point 3. A clear explanation on the varied intramolecular torsional motion under different temperatures should be provided in the abstract, introduction, and conclusion parts, which may induce more consideration of this design and get a larger influence.

Authors's response 3. We have rewritten these sections including special mention to the intramolecular torsional effect. In addition, such as explained in the response to reviewer 1 (point 1) new experimental spectroscopic Raman data have been incorporated in the new version that nicely support all the main argument of the discussion.

Reviewer point 4. About the importance of such a unique phenomenon, the authors are suggested to put more efforts. I think it is not just a new phenomenon, and my feel is that it should be quite meaningful for organic spintronics considering its tunable spin distribution.

Authors's response 4. We agree with this comment. In fact, in the new version of the article we have described the spin-vibration mechanism in the context of the vibronic coupling theory.

Some of these new sentences are added in the Introduction and in the conclusion section of the revised article version.

Reviewer point 5. There are a few typos in the main text and in the information. Please check and correct them. For example, in "... a solid state building scaffold able to bringing intramolecular torsional mobility...", "bringing" should be corrected; In "... bicycle pedal motion in in azobenzene (B)...," please remove one "in".

Authors's response 5. We have checked and corrected grammatical mistakes in the revised manuscripts and the amendments are highlighted in yellow.

REVIEWER #3:

Before addressing the points posed by this reviewer, we would like to draw her/his attention on the new experimental data we provide in the revised version. The data further support our main hypothesis which have allowed to add some more new discussions of the underlying mechanism that overall make a nicer story for the general reader. We hope the referee would agree with us and would be certainly happy if she/he reconsiders his opinion.

Reviewer point 1. ST gap difference between determined values by EPR and SQUID (1.31 vs 3.41 Kcal/mol) is large. One of them must be incorrect.

Authors' response 1. In order to check this, new samples and new SQUID magnetometric measurements have be carried out which result in the data represented in Figure R2 of the response to reviewer 1. By fitting the curves with a modified Bleaney-Bowers equation, the single-triplet energy gap (ΔE_{ST}) now amount to $-3.68 \text{ kcal mol}^{-1}$ which is closer with the result from ESR ($-3.41 \text{ kcal mol}^{-1}$). The new experimental value is also closer to the average value, -2.95 kcal/mol , of the theoretical singlet-triplet gaps data from theory; namely, -4.94 kcal/mol on the crystalline structure at 130K, -1.58 kcal/mol on the crystalline structure at 290K and -2.34 kcal/mol on the crystalline structure at 340K. The new SQUID data have been incorporated to the new version of Figure 3 in the main text together with the discussion of their comparisons.

Figure R2. Temperature-dependent plots of χT versus T and fitted χT - T curve (solid line) for CAR measured at 1.0 T from 2 to 400 K.

Therefore, in pages 12 and 13 of the electronic supplementary information, the new related SQUID data were revised.

In addition, in page 8 line 3 of manuscript, "0.35 emu K mol⁻¹" was replaced with "0.32 emu K mol⁻¹". In Page 8 line 6 of manuscript, "-1.31 emu K mol⁻¹" was replaced by "-3.68 emu K mol⁻¹". And in page 8 line 9 of manuscript, "2-300 K" was replaced with "2-400 K".

Reviewer point 2. ST Thermal driven geometry change of organic diradicals have been reported before, especially by Juan and Wang et al. Those literature work should be clearly described in the text.

Authors' response 2. The referee is right. In fact, the referred work was that cited in reference 15 of our article. It is true that a more in detail explanation of the mechanism reported by Wang in reference 15 was reported by our group years later. In this oligophenyl diradical dication of ref. 15 a normal quinoid→aromatic structural transition in the solid state by heating was observed explained by coupling of the electronic structure with a torsional vibrational mode. In reference 15, the unique reversed effect of the aromatic→pseudoquinoidal conversion by heating was absent, which is now described here for **CAR**. The article cited by the referee is below which has been added to the reference list as reference 30.

Rivero, S. M. et al. Isomerism, Diradical Signature, and Raman Spectroscopy: Underlying Connections in Diamino Oligophenyl Dications. *ChemPhysChem*. 19, 1465-1470 (2018).

Reviewer point 3. STD of geometry parameters should be added in the text and tables.

Authors' response 3. The STD of the bonds and angles were marked in the text and tables in revised manuscript as follows:

In page 5 and in line 2, “38.76°” was replaced with “38.8(2)°”, “5.78°” was replaced with “5.8(5)°”.

In page 5 and in line 3, “1.245 Å” was replaced with “1.245(4) Å”, “1.377 Å” was replaced with “1.377(1) Å”.

In page 5 and in line 7, “1.252 Å” was replaced with “1.252(2) Å”, “1.300 Å” was replaced with “1.300(4) Å”.

In page 5 and in line 11, “1.257 Å” was replaced with “1.257(4) Å”, “1.252 Å” was replaced with “1.252(2) Å”.

In page 6 and in line 1, “1.478 Å” was replaced with “1.478(3) Å”, “1.484 Å” was replaced with “1.484(1) Å”.

In page 6 and in line 10, “1.449 Å” was replaced with “1.449(3) Å”.

In page 7 and in line 6, “5.78°” was replaced with “5.8(2)°”, “8.96°” was replaced with “9.0(3)°”.

In page 7 and in line 7, “8.36°” was replaced with “8.4(4)°”.

Reviewers' Comments:

Reviewer #1:

Remarks to the Author:

I think this revised manuscript is significantly improved and addresses many of my previous concerns. I think that the additional spectroscopic characterization is very helpful. I have only a few minor questions and concerns prior to publication:

1. The Raman data is a very nice addition to the paper, and strengthens the authors' claims substantially. However, it would be nice to show the analysis from Figure 4 top right for all of the vibrational regions. The key trend on going to 340 K is only one data point in both the SXRD data and the Raman data. Is it reproduced in the other Raman features?

2. I just noticed that the TGA data shows significant mass loss upon warming, what is the explanation of this? This is particularly important as the physical phenomena are occurring at 340 K which appears to be in a region where significant mass loss has occurred.

Reviewer #2:

Remarks to the Author:

I believe the paper has been well revised and can be considered for publication in Nat. Commun.

Reviewer #3:

Remarks to the Author:

Shen et al have redone the SQUID and added some previously reported work. Authors also mentioned the disorder of CAR molecule requested by one reviewer. Authors have not completely clarified my questions but just pick up some questions to answer. Based on the provided new data and information, I have to say the conclusion of this manuscript is incorrect.

1. The claimed spin mobility is only based on the crystal structures. Authors now mentioned that CAR is disordered with two conformers especially at high temperatures. The whole molecule is disordered including N-N bond! How can the bond length changes be reliable for a disordered crystal structure?

2. Thus other evidences are badly needed. Authors calculated the spin density directly using the disordered crystal structures and claimed that electron spin density has large changes from low temperatures to high temperatures. Such spin density change needs experimental evidences. Solid state EPR and SQUID should demonstrate this! But they did not! As they states in the text "Unfortunately, these three ΔE ST gaps 270

could not be resolved experimentally by SQUID magnetometry which overall detects one single process", which actually is a strong proof that the bond length changes based on the disordered crystal structures are unreliable! There are no apparent spin density changes with temperature!

3. Authors have not excluded the monoradical impurity that readily exists during synthesis. If we define the parent molecule as CAR-2H, the monoradical is CAR-H and "diradical" is CAR. CAR-H can be buried in the same crystal! The parent molecule CAR-2H can also readily co-crystalize with CAR. In fact, the EPR spectra appear from monoradical CAR-H. Solution NMR and EPR measurements are much better to exclude CAR-2H and CAR-1H.

In conclusion, the purity of claimed CAR molecule is still in doubt. Despite this, CAR is only a normal diradical with a singlet ground state. There is no clear evidence for bond length and spin density changes. This manuscript should not be published in any journal.

REVIEWER 1

Reviewer point 1: The Raman data is a very nice addition to the paper, and strengthens the authors' claims substantially. However, it would be nice to show the analysis from Figure 4 top right for all of the vibrational regions. The key trend on going to 340 K is only one data point in both the SXRD data and the Raman data. Is it reproduced in the other Raman features?

Author's response 1: We have added the representation of the behavior of the areas of of the Raman bands in the 1200-1100 cm^{-1} as a function of the temperature such as it was done for the 1620-1540 cm^{-1} region in Figure 4 (top-right). The new data are included in the same graph (Figure 4) in red colors. The behavior is fully consistent with the previous one. The new Figure 4 is below:

Figure 4. Solid-state variable temperature Raman spectra of **CAR** taken with the 785 (top) and 633 nm (bottom) laser excitations in two different vibrational intervals where the four important (strongest) bands are denoted as (1) $\nu(\text{CC})_{\text{ph}}$; (2) $\nu(\text{CC})_{\text{phO}}$; (3) $\nu(\text{CN})_{\text{azo1}}$ and (4) $\nu(\text{CN})_{\text{azo2}}$. Top, right: representation as a function of temperature of the ratio of intensities of 1 (A1) and 2 (A2) as black circles and 3 (A3) and 4 (A4) bands as red squares as spectroscopic markers of the degree of quinoidal/aromatic character which delineates three regions: small ratio means more quinoidal, large ratio more aromatic and pseudoquinoidal (PQ) is in between. Bottom, right: topologies of the vibrational modes from the theoretical Raman spectra in Supplementary Figs. 12, 13 and 14.

Reviewer point 2: I just noticed that the TGA data shows significant mass loss upon warming, what is the explanation of this? This is particularly important as the physical phenomena are occurring at 340 K which appears to be in a region where significant mass loss has occurred.

Author's response 2: Sorry for both TGA (in $^{\circ}\text{C}$) and Raman/XRD (in K) data being in different temperature scales. According to the TGA in Figure S3 up to approx. 130 $^{\circ}\text{C}$ (approx. 400 K) no significant loss of mass is advised. So, the XRD and Raman data taken at the highest 340 K temperature are in conditions free of decomposition.

REVIEWER 3

Reviewer point 1: Shen et al have redone the SQUID and added some previously reported work. Authors also mentioned the disorder of CAR molecule requested by one reviewer. Authors have not completely clarified my questions but just pick up some questions to answer. Based on the provided new data and information, I have to say the conclusion of this manuscript is incorrect.

Author's response 1: We regret the reviewer does not find our article of interest. We, however, disagree with her/his statement about the incorrect interpretation of the data. We will describe our reasons below. Now, we wish to clarify that we have tried to give responses to all her/his previous questions. So, we did not exercise in selecting some points to positively respond them (ignoring the rest). We would like to remark that we addressed all questions from all reviewers: in some cases, these imply changes in the manuscript and in others no. But always we provide the best responses we can or we are aware of.

Reviewer point 2: The claimed spin mobility is only based on the crystal structures. Authors now mentioned that CAR is disordered with two conformers especially at high temperatures. The whole molecule is disordered including N-N bond! How can the bond length changes be reliable for a disordered crystal structure?

Author's response 2: The bicycle pedal motion is a well-established feature of the crystal of azobenzene, and of its derivatives, and it is well reported in the literature that the way (one of them) to proof this is by seeing some disorder in the azo moiety. This must be remarked since we are dealing with a local effect and it is not a disorder feature of the whole molecule as a function of the temperature. This is clearly stated in the literature [Acta Crystallogr., Sect. B:Struct. Sci., 1997, 53, 662–672] for unresolved disordered structure of azobenzene. The paper concludes: "*the influence caused by the displacement of the positions of the other atoms is so small that it can be negligible which can be proved from their single crystal at 82 K and 296 K*", what is actually our results. The small displacement of the atoms for phenoxy and azobenzene rings is marginal and thus the bond length changes in the rings are reliable and not caused by disorder. Nonetheless, we have to admit that the observed NN bond length might be slightly shorter than the true length owing to the disorder at 250, 290 and 340 K, but this deviation is assumed to be small because the changes of population of the minor conformer is from 10% to 17% on heating while the NN bond lengths show large changes from 1.285(3) to 1.253(4) Å.

So, the presence of disorder in the azo group (insist, not in the whole molecule) is the corroboration of the activation of the bicycle pedal motion which detunes the reversal of the aromatic→quinoidal transition.

The following paragraph has been added to page 6 of the ESI file of the second revision of the article:

In addition, in spite of the presence of the disorder, the population of the minor conformer is very small all along the whole temperature range analyzed even if its ratio is rising with temperature increasing. The very little influence of the disorder on bonds length is mainly existing in the NN bond, but for the other bonds, this is negligible

according to the literature report by: Harada, J.; Ogawa, K.; Tomoda, S. Molecular motion and conformational interconversion of azobenzene in crystals as studied by X-ray diffraction. *Acta Cryst.* **1997**, B53, 662-672.

Finally, we would like to mention that upon requests of reviewers the addition of the variable temperature Raman data considering up to 12 different temperatures in this range corroborate all the behavior deduced from X-ray analysis.

Reviewer point 3: Thus other evidences are badly needed. Authors calculated the spin density directly using the disordered crystal structures and claimed that electron spin density has large changes from low temperatures to high temperatures. Such spin density change needs experimental evidences. Solid state EPR and SQUID should demonstrate this! But they did not! As they states in the text “Unfortunately, these three ΔE_{ST} gaps could not be resolved experimentally by SQUID magnetometry which overall detects one single process”, which actually is a strong proof that the bond length changes based on the disordered crystal structures are unreliable! There are no apparent spin density changes with temperature!

Author’s response 3: We agree that the solid state ESR and SQUID did not show very clear transitions. In fact, If one looks closely on the SQUID data, there is a small transition at 290 K. We have tried to make Bleany-Bowers fits of χT versus T in the temperature ranges lower and higher than 290 K from which we did obtain two ΔE_{s-t} values. However, the error of one of them is relatively large. Therefore, we did not discuss it.

On the other hand, she/he mentioned that other evidences are needed to confirm our hypothesis and these have been provided by variable temperature Raman spectroscopy which further corroborates our arguments. Briefly, Raman spectra have been carried out up to in 12 different temperatures between 130 and 340 K and the spectral changes were analyzed in two spectra regions. In all these temperatures and in the two spectral regions the behavior is fully pointing to the presented hypothesis (see Figure 4 and discussion). So, it is not totally true that we sustain our claims only based on XRD data and only in some selected temperatures. In addition, we reproduce all these features by quantum chemical calculations which are further corroborations of our study. It is a well accepted procedure to obtain the spin distribution from the x-ray data by quantum chemical calculations and this has been done for innumerable cases of magnetic organic molecules.

Reviewer point 4: Authors have not excluded the monoradical impurity that readily exists during synthesis. If we define the parent molecule as CAR-2H, the monoradical is CAR-H and “diradical” is CAR. CAR-H can be buried in the same crystal! The parent molecule CAR-2H can also readily co-crystallize with CAR. In fact, the EPR spectra appear from monoradical CAR-H. Solution NMR and EPR measurements are much better to exclude CAR-2H and CAR-1H.

Author’s response 4:

Thanks for your comment. The oxidation of **CARH** to **CAR** takes place in very high yield (>95%). Also, the polarities of CARH and CAR are very different on silica gel, which can be easily separated by column chromatography. The **CAR** is very stable and was purified from silica gel column (following by thick layer silica plate purification for 2 times). According to our experience on radical purification, monoradical based on phenoxy unit (CAR-1H) is unstable in silica gel. Therefore, all monoradical impurity can

be easily removed. Meanwhile, we store our sample in the glovebox. Furthermore, radical compounds are normally NMR silent, so it is hard to identify their purity based on NMR measurements. We also obtained the crystal structure of **CAR-2H** (**CARH**, Figure 1), the packing mode is very different from that of **CAR**. It will be unlikely that both could co-crystalize. Therefore, we think our sample is very pure enough for all the measurements.

Reviewer point 5: In conclusion, the purity of claimed CAR molecule is still in doubt. Despite this, CAR is only a normal diradical with a singlet ground state. There is no clear evidence for bond length and spin density changes. This manuscript should not be published in any journal.

Author's response 5:

Most of the experimental data presented in the article are scrutinized by the corresponding simulations (XRD, EPR, Raman, singlet-triplet gaps, etc.) and all simulations reproduce very well the experiments. So, assuming there would a fraction of impurity due to precursor radical or others, these seem not to interfere in the analysis and thus in the conclusions. The referee knows well (she/he is a diradical expert) that small fraction of radical impurities in the samples of ALL diradicals are unavoidable. The important thing is that these impurities do not interfere the analysis and this is warranted in our article.

We obviously respect the referee conclusion but are not in agreement. This is not another diradical of the many in the list. CAR diradical represents the unique and first (to the best of our knowledge) example in which molecular vibrations in the ground electronic state modulate the structure and subsequently the spin distribution. So far, most of the existing diradicals are described at their "vibrationally frozen" structure neglecting the role of vibrations. This is logical to some point since the immense majority of diradicals can be divided in two main groups: i) those constructed in rigid or very rigid polycyclic aromatic platforms who vibration-structural relationship is insensitive to rather small-moderate changes of temperature. And ii) those diradicals based on more flexible structures that, unfortunately, are tough to crystalize and become much more reactive. On these flexible diradicals, temperature studies are impractical in the sense that temperature degrades the samples. In this scenario, CAR is in the intermediate situation between the (i) and (ii) cases above in which the effect of temperature in the solid state dynamics can be addressed and this is very significant. In this context, we disagree with CAR being just one more diradical example.

Reviewers' Comments:

Reviewer #1:

Remarks to the Author:

I think the authors have done a good job addressing my, and the other reviewers', concerns. In regard to some of the points raised by the other reviewers, I agree with the concerns raised about drawing conclusions only from the SXR data, particularly at high temperature when disorder is present. However, I think the authors Raman data strengthens their conclusion significantly, and the fact that the same trends are observed across multiple features supports that these effects are real. Furthermore, the timescale of vibrational spectroscopy should be sufficiently fast to avoid similar complications that plague the X-ray data.

As such, I feel that the authors have made their case on an unusual physical phenomenon convincingly and this work should be accepted.

Reviewer #3:

Remarks to the Author:

The reviewer insists that the conclusion of this work is incorrect. 1. The N-N bond length and its change with temperature are key factors in the manuscript. There is no reason to use EADP constraint in the structure refinement because that will considerably affect the N-N bond length. 15% (290K) and 17% (340K) conformers are not trivial at all. If we further consider ESD (around 0.004), the N-N bond lengths (1.253, 290 K vs 1.285, 250 K) do not show any apparent change. 2. Authors still did not exclude the monoradical impurity. 3. No evidences to support the spin mobility, which is only calculated based on unreliable crystal structures. In words, CAR is only a normal diradical with a singlet ground state. There is no clear evidence for bond length and spin density changes. This manuscript should not be published in any journal.

Rebuttal letter for the submission reference **NCOMMS-20-47141B** entitled:
“Normal & Reversed Spin Mobility in a Diradical By Electron-Vibration Coupling”

RESPONSES TO REVIEWER number 3

Reviewer general comment. “The reviewer insists that the conclusion of this work is incorrect.”

Author’s response. Let’s restart the “defense” of our paper by noting a passage of the final evaluation of one of the peers (reviewer number 1) of reviewer number 3 in the evaluation process. In our opinion, this referee has adopted a flexible position in the whole evaluation process that delineates the own evolution of the paper. In the initial round of revision, she/he started by rejecting the article arguing that the conclusions were weakly supported by XRD data alone and she/he asked for new further independent experimental evidences. This was why we conducted Raman measurements and to our delight the data fully supported our claims, upon which the referee reverted her/his decision recognizing the validity of the whole research ending with the recommendation of acceptance. Let’s rewrite part of the passage of this reviewer number 1 in the last round of revision:

“...I agree with the concerns raised by other referee (referee number 3) about drawing conclusions only from the SXRD data, particularly at high temperature when disorder is present. However, I think the authors Raman data strengthens their conclusion significantly, and the fact that the same trends are observed across multiple features supports that these effects are real. Furthermore, the timescale of vibrational spectroscopy should be sufficiently fast to avoid similar complications that plague the X-ray data.”

This passage provides strong arguments clearly explaining why XRD data can be viewed as one of the several information sources on which the paper is based and that there are other techniques, mainly variable temperature Raman spectroscopy, that not only confirm the validity of our hypothesis but also “saves” the conclusions from potentially wrong interpretation raised by the uncertainty of the high temperature XRD data due to disorder. Also, in this line of argumentation, we want to remark that the low temperature XRD data are free of disorder problems. So, the problem resides on two XRD data at 290 and 340 K.

Reviewer comment 1. “The N-N bond length and its change with temperature are key factors in the manuscript. There is no reason to use EADP constraint in the structure refinement because that will considerably affect the N-N bond length. 15% (290K) and 17% (340K) conformers are not trivial at all. If we further consider ESD (around 0.004), the N-N bond lengths (1.253, 290 K vs 1.285, 250 K) do not show any apparent change.”

Author’s response. We have made an additional effort in the attempt to convince reviewer number 3 of our arguments. Figure R1 below shows the comparison of the theoretical Raman spectrum calculated on the molecular geometry taken from the XRD crystalline structures at 290 K and the experimental Raman spectrum taken with the 1064 nm Raman excitation, or FT-Raman spectrum. We take this 1064 nm Raman spectrum (instead of the 633 or 785 nm Raman spectra discussed in other sections) because it is non-resonant with the strongest electronic absorption bands of **CAR** and it is thus closer to the theoretical spectrum which does not consider resonant Raman

effects, neither. We observe that the comparison of both experimental and theoretical spectra is certainly good and all main features with their relative intensities among groups of bands are predicted and recorded quite satisfactorily in both spectra. This reveals that the XRD crystalline structures (at least at 290 K) are good molecular structures to reproduce the highly structurally sensitive vibrational Raman spectrum and consequently supports the discussion of the molecular geometries from XRD at high temperatures (ruling out severe distortions of these geometries due to disorder, at least at 290K). Let's emphasize that vibrational frequencies come from the second derivatives of the potential energy surface of the ground electronic state regarding each normal mode, so, the set of vibrational frequencies (Raman in our case) are highly sensitive to the geometries (in order words, even small geometry deviations would produce large frequency mismatches). Our conclusion with this is that the geometric data from XRD at 290 K are sufficiently good to reproduce the vibrational spectrum and hence should be considered of quality for geometric discussion. We have added this experimental and theoretical comparison to Figure S14 in the new version of the electronic supplementary information file together with a small mention in page 12 of the main text of the article.

Figure R1 (new graph of Fig S14). Full range comparison of (a) theoretical Raman spectrum of **CAR** at the DFT/ B3LYP/6-31G* level calculated on the molecular geometry from the x-ray structure at 290 K and (b) 1064 nm FT-Raman spectrum of **CAR** at 298 K in the solid state. For the comparison of the whole spectrum the 1064 nm FT-Raman spectrum is preferred over the 633 and 785 nm since the former is in non-resonant conditions regarding the strong electronic absorption bands. On non-resonant Raman conditions, a richer vibrational Raman spectrum is obtained that better compares with the theoretical spectrum simulated in non-resonant conditions as well. Shaded band colors allow to correlate the different groups of bands between the spectra.

Reviewer comment 2. “No evidences to support the spin mobility, which is only calculated based on unreliable crystal structures.”

Author's response. From the whole set of Raman data at different temperatures, our spectroscopic analysis reveals the evolution of the electronic structure from a quinoidal to a more aromatic shape in the low temperature range followed by a transition from the aromatic structure to a pseudo-quinoidal one on increasing temperature (i.e., in the high

temperature range). In this class of para-quinodimethane kekulé delocalized diradicals, the stabilization of the unpaired electrons and of their associated spin density in the quinoidal (pseudo-quinoidal) and aromatic forms takes place always along the main π -conjugation path. That is, the bonding and exchange repulsion competition between the unpaired electrons tunes the electronic structure and produces a modulation and change of the delocalization extension (i.e., unpaired electron separation) along the main π -conjugated path among the quinoidal, aromatic and pseudo-quinoidal forms. Associated with the tuning of the wavefunction delocalization extension of the unpaired electrons, there is an equivalent modulation of the spin density and this is what we refer as spin mobility which is activated thermally. So, the discussion in the article of the spin mobility effect is also based on the evolution of the electronic structures deduced from the Raman data and is not “entirely” based on XRD data. Reviewer number 3 supports his strong criticism in the fact that we extract spin densities from XRD data thus plagued with uncertainty. Against this reviewer interpretation, we wish to remark two aspects: i) the fact that the XRD geometries (at least those at 130 K and 290 K) are seemingly fine as discussed from the experimental and theoretical Raman comparison in Figure R1; and ii) in the fact that fully independently from the XRD data we arrive to the same conclusion based on Raman data.

Reviewer comment 3. “Authors still did not exclude the monoradical impurity”.

Author’s response. Regarding objection of referee 3 in regard of the presence of radical impurities, we would like to remark that the presence of these impurities in the final product of most of the reported diradicals is rather unavoidable. It is well assumed in the community of scientists working in organic diradicals that this impurity will be always there and that the effort in the purification of the sample should go to keep it below 1-2% as obtained by SQUID or EPR data. We would like to remark again that our impurity value due to radical as obtained by EPR data is 1.7%. We have put this value in the main text of the new version in page 9 (previously was in the ESI file) to avoid reader confusion in this regard.

Reviewer comment 4. “In words, CAR is only a normal diradical with a singlet ground state. There is no clear evidence for bond length and spin density changes. This manuscript should not be published in any journal”.

Author’s response. Finally, we regret the referee finds **CAR** as an incremental diradical. We respectfully think that this is not the case. **CAR** diradical represents the unique and first (to the best of our knowledge) example in which molecular vibrations modulate the structure and subsequently the spin distribution in a diradical. So far, most of the existing diradicals are described at their “vibrationally frozen” structures neglecting the role of nuclei dynamics. Thus, this is a new appealing novelty of our study that makes it different from others. The reviewer would agree with us in the division of the existing diradicals in two main groups: i) those constructed in rigid or very rigid polycyclic aromatic platforms where the whole electronic structure is insensitive to the thermal population of low frequency vibrational modes and hence rather insensitive to the nuclei dynamics in a small-moderate range of temperature. And ii) those diradicals based on flexible structures that, unfortunately, are tough to crystalize and are usually more reactive than those of group (i) given the difficulty in designing protecting groups of the radical centers that could efficiently work against structural flexibility. As a result, solid

state temperature studies in flexible diradicals are impractical in the sense that their solid-state structures are unknown and temperature easily degrades them. In this scenario, **CAR** is an intermediate situation between (i) and (ii) above in which its solid-state structure is fully solved and the effect of temperature in the solid-state dynamics can be addressed. To our delight, the solid-state nuclei dynamic impact in the electronic structure is unexpected and is associated to a unique pedal motion of the azo group that allows us to uncover an unique electron-vibration mechanism which is certainly unprecedented in the field of organic neutral diradicals. All these reasons, in our opinion, are strong evidences of the novelty of this **CAR** study and its breakthrough nature.

In the current third revised version of the article we have slightly changed the previous text such as previously described. Changes are highlighted in blue in the main text and in the ESI file.

Reviewers' Comments:

Reviewer #1:

Remarks to the Author:

While I appreciate the other reviewer's concerns, I find the authors counterarguments compelling. I think that the XRD data, while not entirely conclusive, do suggest a trend which the authors have interpreted. The Raman and other data further validate the authors interpretation. I think the concerns about $S = 1/2$ impurities have been addressed by the reviewers in as thorough a manner as may be expected. Finally, I think the concerns over terminology such as "spin mobility" are somewhat semantic. I personally don't think of this as spin mobility, but I understand the context the authors use this phrase in.

Overall, I think the sum of the data is consistent with the authors' interpretation. As such, I recommend that the paper be accepted for publication in it's current form.